# Contraception and Hormone Replacement Therapy in Healthy Carriers of Germline BRCA1/2 Genes Pathogenic Variants: Results from an Italian Survey

**DOI:** 10.3390/cancers14143457

**Published:** 2022-07-15

**Authors:** Claudia Massarotti, Barbara Buonomo, Miriam Dellino, Maria Campanella, Cristofaro De Stefano, Alberta Ferrari, Paola Anserini, Matteo Lambertini, Fedro A. Peccatori

**Affiliations:** 1Department of Neurosciences, Rehabilitation, Ophthalmology, Genetics, Maternal and Child Health (DiNOGMI), School of Medicine, University of Genova, 16126 Genova, Italy; 2Academic Unit of Obstetrics and Gynaecology, IRCCS Ospedale Policlinico San Martino, 16132 Genova, Italy; 3Gynecologic Oncology Program, European Institute of Oncology IRCCS, 20141 Milan, Italy; barbara.buonomo@santagostino.it (B.B.); fedro.peccatori@ieo.it (F.A.P.); 4Clinic of Obstetrics and Gynecology, “San Paolo” Hospital, ASL Bari, 70123 Bari, Italy; miriam.dellino@uniba.it; 5aBRCAdabra, National Patient Advocacy Association for Carriers of BRCA Genes Mutation, 20019 Milan, Italy; presidente@abrcadabra.it; 6Department of Women’s and Children’s Health, “San Giuseppe Moscati” Hospital, 83100 Avellino, Italy; fisiopat@aosgmoscati.av.it; 7Department of Surgical Sciences, General Surgery 3—Breast Surgery, Fondazione IRCCS Policlinico San Matteo, 27100 Pavia, Italy; a.ferrari@smatteo.pv.it; 8Physiopathology of Human Reproduction Unit, IRCCS Ospedale Policlinico San Martino, 16132 Genova, Italy; paola.anserini@hsanmartino.it; 9Department of Internal Medicine and Medical Specialties (DiMI), School of Medicine, University of Genova, 16132 Genova, Italy; matteo.lambertini@unige.it; 10Department of Medical Oncology, U.O. Clinica di Oncologia Medica, IRCCS Ospedale Policlinico San Martino, 16132 Genova, Italy

**Keywords:** contraception, menopause, BRCA, HRT

## Abstract

**Simple Summary:**

Being a healthy carrier of a *BRCA1/2* pathogenetic variant is not a contraindication to hormonal contraception and menopause hormonal therapy; however, insufficient knowledge on the topic frequently translates into suboptimal counseling and care. The results of this nationwide Italian survey show that, after being diagnosed as healthy carriers, only 24.5% used hormonal contraception and 28.4% menopause therapy, even though this reduced their quality of life, and the majority were not satisfied with the counseling received. Several misconceptions on the topic persisted, for example, 58.2% were not aware of the protective effect of hormonal contraception on the risk of ovarian cancer. These results highlight the need for educational initiatives on the topic, directed to both healthcare professionals and the population.

**Abstract:**

Several myths and misconceptions exist about hormones in women with familial predisposition to cancer, and there are few real-life data on their prescription and uptake. To better understand how they are prescribed and accepted in healthy carriers of a *BRCA1/2* pathogenetic variant, an online survey was uploaded on Google Forms and shared through social media closed groups of patients’ associations, aBRCAcadabra and ACTO Campania. A total of 241 questionnaires were collected. Sexual quality of life was considered of the utmost importance by most of the respondents (mean score of 7 ± 2.8/10), but they felt the counseling they received by healthcare professionals on the topic was insufficient (4.9 ± 3.2/10). Only 57 women out of 233 (24.5%) had used hormonal contraception after being diagnosed as carriers of a *BRCA* pathogenetic variant, and 42 out of 148 (28.4%) underwent menopause hormonal therapy. The majority of women (53.6% for contraception and 61.5% for menopause) reported being dissatisfied with the counseling received, and 58.2% were not aware of the protective effect of hormonal contraception on the risk of ovarian cancer. An educational effort is desirable to guarantee healthy BRCA carriers reliable contraception and evidence-based menopause counseling.

## 1. Introduction

Pathogenic variants in the *BRCA 1/2* genes represent a significant risk factor for cancer development, with a lifetime cumulative risk up to 72–69% for breast cancer and 44–17% for ovarian cancer for *BRCA1* and *BRCA2* pathogenic variant carriers, respectively [1]. The estimated prevalence of *BRCA1* and *BRCA2* pathogenic variants in the general population is between 1 in 300 and 1 in 800 [2]. As a consequence of extensive BRCA screening in breast cancer, ovarian cancer, pancreatic cancer, and prostate cancer patients, more women are diagnosed as healthy carriers and have the chance to undergo risk-reducing surgery, as indicated by guidelines [2,3,4]. At the time of these procedures, healthy *BRCA* carriers are usually young. Therefore, they face several decisions on gynecology-related issues, including contraception, fertility, and pregnancy [5]. Furthermore, undergoing risk-reducing salpingo-oophorectomy (RRSO) before the age of 40 for *BRCA1* and 45 years for *BRCA2* carriers is associated with premature iatrogenic menopause.

Both hormonal contraception and hormonal replacement therapy (HRT) after menopause are not contraindicated in healthy *BRCA* carriers [6,7]. On the contrary, there is evidence that combined hormonal contraception decreases ovarian cancer risk both in the general population and in healthy *BRCA* pathogenic variant carriers [8]. The guidelines on premature menopause suggest HRT up to 50 years, not only to manage immediate adverse effects such as vasomotor or genitourinary symptoms, but also to prevent long-term consequences on bone health, the cardiovascular system, and cognitive function [9]. Indeed, there is evidence in the general population that mortality is significantly higher in women who had received RRSO before the age of 45 and did not undergo HRT (hazard ratio 1.67 (95% CI 1.16–2.40), *p* = 0.006) [10]. In *BRCA* carriers, RRSO has been shown to decrease gynecologic cancer-specific, as well as overall, mortality [11], but the short- and long-term adverse effects of premature menopause still remain. A recent meta-analysis did not find increased cancer risk linked to the use of HRT in healthy carriers, which is not contraindicated in this specific population [12].

However, there are few data on real-life uptake of contraception and HRT in *BRCA* healthy carriers. Studies from small cohorts report that 40 to 60% of women receive HRT after prophylactic RRSO [13,14], while we have no data on how hormonal contraception is prescribed and accepted after detection of a germline pathogenic variant in the BRCA genes.

## 2. Materials and Methods

On 5–7 December 2019, patients’ advocates and physicians with expertise in the field of reproductive medicine, fertility preservation, and oncology were invited to “San Giuseppe Moscati” Hospital in Avellino (Italy) to participate in a workshop on the reproductive management of women with germline pathogenic variants in the BRCA1/2 genes [5]. The invited experts represented different disciplines related to the topic, including oncologists, gynecologists, geneticists, and bioethicists. Starting from patients’ needs voiced by the advocates, several issues were discussed—among them, the common perception of a diffuse suboptimal knowledge about hormonal contraception and HRT. To better assess women’s attitudes and to plan educational measures to improve quality of care, a nation-wide survey was launched aiming to collect information on the experience of healthy BRCA carriers with the use of hormonal contraception and HRT. Patients with a history of cancer were excluded. After informed consent, respondents were asked to answer two different modules: one about contraception after diagnosis of being healthy carriers and one about menopause (see Appendix A for an English translation of the survey). There was the possibility to leave one module blank: if the woman, during her fertile years, was not aware of having a BRCA pathogenic variant, she was asked to leave the contraception module blank; if she was not postmenopausal yet, she was asked to leave the menopause module blank. The survey was disseminated by aBRCAcadabra and ACTO, two Italian patients’ associations representing BRCA mutation carriers and ovarian cancer patients. Members were contacted through the associations’ private groups on social media (2000 and 600 members, respectively, of which approximately 400 and 50 were BRCA carriers who had never had a cancer diagnosis) and invited to fill out the questionnaire through Google Forms in January and February 2021. Since the survey was anonymous and the subjects thus cannot be identified, either directly or through identifiers, it was considered IRB-exempt. All respondents agreed to a disclaimer explaining the use of the information they volunteered (see Appendix A). The response rate was calculated with AAPOR Outcome Rate Calculator spreadsheet, Version 3.1 November 2010 (AAPOR, Alexandria, VA, US).

Continuous data were descriptively reported as mean ± standard deviation and compared through Student’s T test, and categorical data as absolute numbers and percentages compared through χ2- test. Univariate and then multivariate logistic regression were used to study possible predictors of suboptimal contraception/menopause usage. The R software, Version 3.6.2, was used for statistical analyses. A *p*-value <0.05 was considered significant.

## 3. Results

### 3.1. Study Population

A total of 241 questionnaires were collected, with a response rate of 0.535 (54% of the eligible women). Among them, five were excluded from analyses: one for the absence of informed consent (left blank) and four because the questionnaires were incomplete or left entirely blank. The characteristics of the remaining 236 study participants are reported in Table 1. 

The mean age was 40 ± 8.6 years (range: 18–62), with a reported mean age at BRCA pathogenic variant diagnosis of 35 ± 8.8 years (range: 17–56). A total of 137 (58.1%) women carried a pathogenic variant of the BRCA1 and 99 (41.9%) of the BRCA2 gene. The mean time from carrier status detection to survey was 5 ± 4.3 years (range 0–23). 

In total, 174 women (73.7%) had undergone risk-reducing surgery (43, 18.2% bilateral mastectomy; 42, 17.8% RRSO; and 89, 37.7% both) at the time of the survey. Women who underwent risk-reducing surgery were older (43.2 ± 7.5 versus 34.2 ± 7.8 years, *p* < 0.001) and were known to be carriers of a pathogenic BRCA1/2 variant for a longer period (5.4 ± 4.7 vs. 4.1 ± 3.4 years, *p* = 0.04). 

When asked to give a score from 0 (= not important at all) to 10 (= of utmost importance) to their quality of sexual life, study participants answered with a mean score of 7 ± 2.8, the most frequent answer being 10/10 (57 women, 24.2%). However, when asked how much attention was given to this aspect by medical professionals, the mean score dropped to 4.9 ± 3.2, with the most frequent answer being 0/10 (35 women, 14.9%).

### 3.2. Contraception

Out of 236 women, 233 (98.7%) filled out the contraception module. A total of 125 (53.6%) women declared not to be satisfied with the information received on the topic. After counseling with any physician, 52 (22.3%) declared to be more worried than before about the possible cancer risks linked to contraception. A total of 130 (55.8%) women declared to have never discussed the topic specifically with a gynecologist: 94 (72.3%) of them because they did not need hormonal contraception, and 36 (27.7%) women because they thought they could not use it as healthy BRCA carriers. Out of the other women, 46 (19.7%) discussed contraception prompted by the gynecologist and 49 (21%) because they proactively asked. 

Among the 57 (24.5%) women reporting hormonal contraception use, 46 (19.7%) underwent short-acting contraception with estrogens and progestins, 6 (2.5%) with a levonorgestrel intrauterine system (IUS), and 5 (2.1%) with a progestin-only pill. Among the other women, 5 (2.1%) opted for a copper intrauterine device (IUD), 10 (4.3%) decided for RRSO right after carrier status detection, 2 (0.8%) abstained from intercourse, 45 (19.3%) used a barrier method, and 8 (3.4%) did not answer. The remaining 106 women (45.5%) explicitly stated they did not use any contraceptive method (Figure 1A). 

When asked about the oncological risk related to BRCA1/2 pathogenic variants, 135 (65.5%) women answered that hormonal contraception increases the risk of breast cancer and 23 (11.2%) that it increases the risk of ovarian cancer (Figure 2A).

Younger age (OR 0.93, 95% C.I. 0.89–0.97, *p* = 0.001) and having been reassured by the gynecologist during a dedicated counseling (OR 10.80, 95% C.I. 4.61–23.68, *p* < 0.001) were predictors of hormonal contraception usage in a multivariate model. Instead, having underwent prophylactic mastectomy was not a predictor of hormonal contraception usage (OR 0.76, 95% C.I. 0.37–1.55, *p* = 0.45). 

### 3.3. Menopause

One hundred forty-eight (66.9%) women filled out the menopause section. The majority of them (*n* = 123, 83.1%) declared that menopause was consequent to risk-reducing surgery. The mean age of menopause was 42.15 ± 4.67 years. Ninety-one women (61.5%) reported not having had enough information from healthcare providers on the adverse effects of premature menopause and possible therapies. Half of the respondents (*n* = 78, 52.7%) discussed menopause-related issues with a gynecologist because they asked (*n* = 41, 25%) or were prompted by the doctor (*n* = 37, 27.7%). When asked whether they discussed these issues with any physician, the number of women who answered yes increased to 107 (72.9%). Only 27 women out of 107 (25.2%) reported having felt reassured about HRT safety after counseling. The majority of women never used HRT (97, 65.5%). Only 25 women (16.9%) underwent systemic HRT with estrogens and a progestin (E+P HRT), 10 (6.8%) used a combination of conjugated equine estrogens and bazedoxifene, 7 (4.7%) tibolone, 2 (1.4%) topical estrogens, and 7 (4.7%) did not answer (Figure 1B). As for alternative remedies, 18 women reported phytoestrogen use (12.4%), and 40 used non-hormonal vaginal moisturizers and lubricants (27.6%); 73 women changed their diet and exercise habits (50.3%), and 31 used dietary supplements such as vitamins and minerals (21.4%).

When asked about symptoms consequent to postmenopausal low estrogen levels, 88 (59.5%) women declared to be highly symptomatic (vasomotor symptoms, genitourinary syndrome, mood swings, or all of them), 17 (11.5%) declared to be interested in HRT for preventing the long-term consequences of premature menopause (bone health, cardio-vascular diseases, cognitive function), 25 (17%) did not answer, and only 18 (12.1%) reported no symptoms or no interest in the possibility of HRT. 

To better explore the reason behind the gap between the high prevalence of symptoms and the low treatment uptake, we asked whether they believed that HRT had an effect on BRCA1/2-related cancer risk. A total of 99 (62.8%) women answered that it increases breast cancer risk and 38 (25.7%) ovarian cancer risk (Figure 2B). However, 72.3% of the respondents answered that this specific risk was not asked about or addressed after risk-reducing surgery. Out of the 103 women who answered about why they did not take HRT, 16 (15.5%) said because they would not need it, while 42 (40.8%) refused HRT because they were worried about the potential cancer risk consequent to hormonal therapies; 41 (39.8%) answered that the doctor refused to prescribe it, and 4 (3.9%) preferred not to disclose the reason. 

Among women who answered that the doctor refused HRT, the mean age was 44.8 ± 8.4 years, the mean age at menopause was 41.4 ± 4.7 years, and all but two of them were menopausal because of an RRSO. The most frequent symptom was genitourinary syndrome. When asked to give a score from 0 (worse possible) to 10 (best possible) to their sexual life, the mean was 3.4 ± 2.9, with the most frequent answer being 0 and only nine scores equal or superior to 6.

A previous risk-reducing mastectomy was a predictor of HRT use (OR 2.87, 95% C.I. 1.24–7.32, *p* = 0.007). Age and type of menopause (physiological vs. surgical) were not predictive of HRT usage (OR 1.0, 95% C.I. 0.93–1.07, *p* = 0.90 and OR = 3.75, 95% C.I. 0.89–26.1, *p* = 0.11, respectively).

Among the women who underwent prophylactic double mastectomy, 93 were postmenopausal and filled out the “menopause” module of the questionnaire. Their mean age was 41.60 ± 4.04 years old; all but eight underwent surgical menopause from RRSO. The use of HRT was higher than in postmenopausal women who underwent mastectomy (38.7% vs. 16.3%, *p* = 0.006), and in both subgroups, the most used therapy was systemic HRT containing estrogens and progestins (21/36 women after mastectomy, 58.3%, and 4/8 women without mastectomy, 50%, *p* = 0.66). The women who underwent mastectomy rated their satisfaction with the HRT counseling received 4.83 ± 2.89 and the quality of their sexual life 4.83 ± 2.89 on a scale from 1 to 10. Similarly, those who did not undergo mastectomy rated the HRT counseling received and the quality of their postmenopausal sexual life 3.94 ± 2.99 and 4.79 ± 3.07, respectively. 

## 4. Discussion

This is the first survey reporting the uptake of hormonal contraception and HRT in healthy *BRCA1/2* carriers in Italy. The results show a low uptake of hormonal methods, even if not contraindicated, and a general dissatisfaction with the counseling received, underlining the need for educational initiatives for both physicians and patients regarding this topic. 

A suboptimal usage of highly effective contraception exposes women to the risk of unwanted pregnancies. Moreover, combined hormonal contraception has a risk-reducing effect on ovarian cancer with an inverse correlation with its duration [15]. This beneficial effect was proven specifically in *BRCA1/2* carriers by numerous case–control cohort studies and four meta-analyses [8]. Moorman and colleagues performed a meta-analysis of one cohort study (3181 participants) and three case–control studies (1096 cases and 2878 controls) to explore whether the use of hormonal contraception predicted ovarian and breast cancer occurrence. The authors found a significant reduction in ovarian cancer risk and no increase in breast cancer risk for both *BRCA1* and *BRCA2* carriers [16]. Nonetheless, among those responding to our survey, only 29.1% and 41.8% knew of hormonal contraception safety for breast and ovarian cancer, respectively. Moreover, consulting a physician was not resolutive in many cases. 

As for HRT prescription and uptake, the situation depicted by our results was even worse. The respondents reported having invalidating symptoms (such as genitourinary syndrome keeping them from having a regular sexual life) that were not treated. In 39.8% of the cases, this could be explained by a strong refusal from the physician for the fear of a potential increased cancer risk, especially in women who did not undergo risk-reducing mastectomy. However, the existing literature does not suggest an increase in breast cancer risk in the specific cohort of *BRCA1/2* healthy carriers undergoing HRT, even before breast surgery [13,17,18]. A recent meta-analysis that included three studies and a total of 1100 *BRCA* healthy carriers found no HRT-correlated increased risk of breast cancer (RR = 1.01; 95% CI = 0.16–1.54) [19]. The type of HRT may impact breast cancer risk: we know from the general population that estrogen-only HRT is associated with a reduced risk of breast cancer and should be considered the first choice for women who have undergone hysterectomy [20]. The PROSE study reported no alteration in the breast cancer risk reduction associated with RRSO with the use of HRT; however, the majority of patients used estrogens alone rather than combined with a progestin (58% versus 22%) [18]. The prospective longitudinal study by Kotsopoulos included 377 *BRCA1* healthy carriers who used HRT after RRSO and 495 who did not. This study reported a risk reduction correlated with estrogen-only HRT and a non-significant increase in breast cancer risk with HRT including estrogens and a progestin [17]. HRT after RRSO was considered safe, including before risk-reducing mastectomy, with the recommendation of using estrogen-only HRT in patients without a uterus and to minimize systemic progestin exposure in the others using a progestin-containing IUS [21,22]. Moreover, vaginal estrogens, highly efficacious for managing genitourinary syndrome from menopause [20], have no contraindications in healthy *BRCA* mutation carriers, and yet their use in our sample was very limited as well (1.4% of patients). 

Many respondents declared to have undergone RRSO (55.5%) and prophylactic double mastectomy (55.9%), a slightly higher percentage than the 38% and 40% reported in the literature [11,23]. Aside from the full reimbursement of the procedure by the public healthcare system in Italy, this could also be explained by the fact that they belong to patients’ associations, a network of mostly more knowledgeable and self-aware women. We hypothesize that the results would have probably been even worse in the general population. 

The survey was taken among the Italian population, where we know several myths and misconceptions about hormonal methods persist. For example, the 2020 European Contraception Policy Atlas reports that only 59.3% of Italian women of reproductive age use contraception, with Italy being 23rd in Europe for contraception uptake [24]. The uptake of HRT after physiological menopause is also low in Italy (less than 20% of women) [25], despite its being fully reimbursable by the Italian healthcare system, but its usage significantly increases in cases of premature menopause under the age of 45, as it is recommended by guidelines [9].

Other than a low uptake, the survey showed relevant safety misconceptions specifically correlated with the *BRCA* carrier status, frequently shared by healthcare providers. 

## 5. Conclusions

The situation depicted by our results is worrisome and mandates a call to action for better education in this area. While increasing attention is being given to *BRCA1/2* healthy carriers regarding risk-reducing surgery for cancer prevention, the safety of pregnancy, and the possibility of fertility preservation, it is also crucial to highlight the importance of reliable contraception and evidence-based HRT counseling.

## Figures and Tables

**Figure 1 cancers-14-03457-f001:**
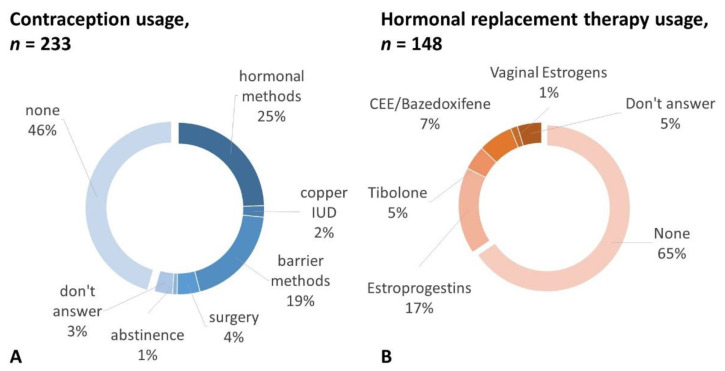
Contraception (**A**) and hormonal replacement therapy (**B**) self-reported use in healthy *BRCA 1/2* carriers. IUD = intrauterine device; CEE = conjugated equine estrogens.

**Figure 2 cancers-14-03457-f002:**
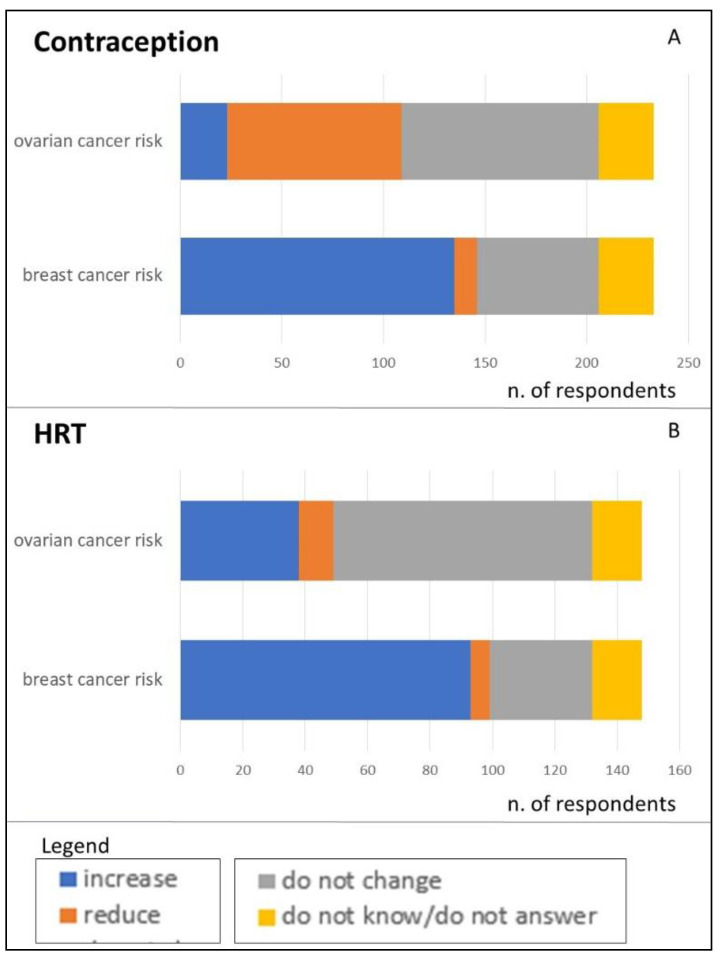
Beliefs of healthy BRCA 1/2 carriers regarding the oncological risk linked to use of contraception (**A**) and hormone replacement therapy (HRT) (**B**).

**Table 1 cancers-14-03457-t001:** Demographic characteristics of study participants.

**Age**, years	40.8 ± 8.6 (18–62)
**Education**, *n* (%)	
< High school	16 (6.8)
High school degree	90 (38.1)
University or higher	130 (55.1)
**Having children**, *n* (%)	
Yes	163 (69.1)
No	73 (30.9)
**Age at healthy carrier status diagnosis**, years	35.8 ± 8.8 (17–56)
**BRCA1**, *n*. (%)	137 (58.1)
**BRCA2**, *n*. (%)	99 (41.9)
**Region**, *n*. (%)	
North Italy	124 (52.5)
Central Italy	59 (25)
South Italy	36 (15.3)
Islands	17 (7.2)
**Access to a fertility unit**, *n*. (%)	
Yes	124 (54.5)
No	83 (35.2)
Don’t know/no answer	29 (12.3)
**Risk-reducing surgery**, *n*. (%)	
Mastectomy only	43 (18.2)
RRSO only	42 (17.8)
Both	89 (37.7)
None	62 (26.3)
**Age at mastectomy**, years	39.6 ± 6.8 (22–57)
**Age at RRSO**, years	42.9 ± 5.5 (28–57)

Continuous data are expressed as mean ± SD (min–max), and categorical data are expressed as number (%). RRSO: risk-reducing salpingo-oophorectomy.

## Data Availability

Data are available upon reasonable request.

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
