# Peer review of "Contraception and Hormone Replacement Therapy in Healthy Carriers of Germline BRCA1/2 Genes Pathogenic Variants: Results from an Italian Survey"

_cancers, 2022, doi:10.3390/cancers14143457_

Round 1

Reviewer 1 Report

The topic of the manuscript is of interest for the management of young BRCA mutation carriers undergoing preventive surgery.  There are some items that could improve the manuscript summarized below:

1. IRB exemption -- is this true for your institution?  typically any patient facing contact must receive REB approval.  please explain.

2. Introduction

- authors should consider elaborating on impact of oophorectomy on outcomes in carriers also

-any insight into uptake of surgery in carriers in Italy?  

- are there recent analyses that have been conducted in other countries? should integrate

3. Methods

- grammatical error on line 92 ...please rephrase as it is not clear as it stands

4. Results

- a key issue with the analysis and presentation of findings is that the authors did not compare hormonal exposure among mastectomy or oophorectomy groups separately

- importantly, are they women all unaffected by disease?  this needs to be discussed as indications differ for those with a personal history of disease

- Figures -- very hard to read and require more information in the legends.

-several contraceptives are queried, however, some more discussion of bazedoxifene [which is incorrectly spelled in this paper].  this is a less commonly used drug with a lot of potential

- Figure 2 and all the results should be stratified by surgical group.

5. Conclusion

- check spelling/grammar lines 224 and 264

- the conclusion needs some more integration of existing data on the topic from other countries as well, private vs. public health care impact, etc.

Author Response

We would like to thank the Reviewer for the raised comments, which we believe have significantly improved the quality of our manuscript. Kindly find below our responses to the raised comments. In the uploaded revised manuscript, the changes are highlighted in yellow. We hope that we were able to respond adequately to the raised comments. 

Reviewer #1:

The topic of the manuscript is of interest for the management of young BRCA mutation carriers undergoing preventive surgery.  There are some items that could improve the manuscript summarized below:

  1. IRB exemption -- is this true for your institution? typically any patient facing contact must receive REB approval. please explain.

Reply (R) 1. As explained in Methods and in the questionnaire itself (you can read the translated version as “supplementary material 1”), the survey was completely anonymous, without collection of any identifying information (name, email, IP address). All patients ticked “yes” to a disclaimer agreeing to the use of the data, in aggregate form, for research and dissemination purposes (this was ticked and not signed, because we were not collecting any identifying information). Anonymous surveys qualify as exempt for IRB.

  1. Introduction - authors should consider elaborating on impact of oophorectomy on outcomes in carriers also

R2: As the Reviewer suggested we added two sentences and two references to elaborate on the impact of oophorectomy specifically in carriers (see page 2, line 71-75 and references n 11-12). Moreover, the concept is expressed in the Discussion section as well.

-any insight into uptake of surgery in carriers in Italy? Are there recent analyses that have been conducted in other countries? should integrate

R3: as suggested by the reviewer we integrated the data with what is known about the uptake of prophylactic surgery for BRCA carriers (see Discussion, page 8, line 278-292)

  1. Methods - grammatical error on line 92 ...please rephrase as it is not clear as it stands

R4: We thank the Reviewer for this observation. We have rephrased the sentence for clarity (see Methods, page 2-3, line 96-99)

  1. Results - a key issue with the analysis and presentation of findings is that the authors did not compare hormonal exposure among mastectomy or oophorectomy groups separately

R5: previous prophylactic double mastectomy or RRSO were evaluated as possible predictors of contraception and HRT usage in the multivariate analysis: the only significative finding was that a previous mastectomy was predictive of HRT usage (as it can be read in the Results section). We, however, understand that it may have not been stated clearly in the manuscript so we modified results as follow: 
- in the Contraception paragraph of the Results section, we added how prophylactic mastectomy was not a predictor of hormonal contraception usage (we added: “Instead, having underwent prophylactic mastectomy was not a predictor of hormonal contraception usage (OR 0.76, 95% C.I. 0.37-1.55, p=0.45)”, see Results, page 7 line 177-179).
- we explored if type of menopause (RRSO vs. physiological) could be predictive of HRT usage and the results are not statistically significant. Following the Reviewer’s suggestion, we reported the results in the Menopause paragraph of Results (see Results, page 7, line 220-223)
- since prophylactic mastectomy was found predictive of HRT usage, we added a paragraph in results comparing HRT usage, type of HRT, satisfaction with the counselling received and sexual quality of life in post-menopausal women who underwent a prophylactic mastectomy to the same data in those who didn’t. The results are reported at page 7 and 8, line 220-234.

- importantly, are they women all unaffected by disease?  this needs to be discussed as indications differ for those with a personal history of disease

R6: this is a survey specifically aimed to healthy carriers and only them, as it is said in the title and in the manuscript. The indications differ significantly in patients who have had cancer and they are not the subject of our paper. Following the suggestion of the reviewer, we further clarified it (see Methods, page 2, line 92-93)

- Figures -- very hard to read and require more information in the legends.

R7: following the suggestion of the Reviewer we modified the figures to improve clarity and added more information (including acronyms) in the legends.

-several contraceptives are queried, however, some more discussion of bazedoxifene [which is incorrectly spelled in this paper].  this is a less commonly used drug with a lot of potential

R8: We corrected the spelling error in the manuscript and in Figure 1. In Italy Bazedoxifene is approved to be marketed only together with conjugated equine estrogens and it is currently facing production issues, so it is very hard to find. This is probably the reason of the low uptake. We agree that it has potential, but right now we wanted to provide a snapshot of the reality, rather than review other possibilities, even if promising.

- Figure 2 and all the results should be stratified by surgical group.

R9: In the questionnaire we asked patients how hormonal contraception and HRT could impact breast and ovarian cancer risk in healthy carriers of BRCA1/2 pathogenic variants, independently to prophylactic surgery and their specific situation (otherwise the variable would have been too many and the numerosity insufficient).  

The questions were the following:

“The use of hormonal contraception in BRCA pathogenic variant carriers that never had a cancer:

Increases ovarian cancer risk

Decreases ovarian cancer risk

Does not alter ovarian cancer risk

Increases breast cancer risk

Decreases breast cancer risk

Does not alter breast cancer risk”

and

“The use of HRT in BRCA pathogenic variant carriers that never had a cancer:

Increases ovarian cancer risk

Decreases ovarian cancer risk

Does not alter ovarian cancer risk

Increases breast cancer risk

Decreases breast cancer risk

Does not alter breast cancer risk”

We were in fact NOT asking “what is your specific risk?”. That would have had numerous confounders (including familial history, age, etc). The questions are therefore not formulated to be analyzed separately depending on prophylactic surgery.

  1. Conclusion - check spelling/grammar lines 224 and 264

R10: we thank the Reviewer, we rephrased the sentences to correct grammar/spelling (see Discussion, page 8, line 253 and Conclusions page 9, line 297-299)

- the conclusion needs some more integration of existing data on the topic from other countries as well, private vs. public health care impact, etc.

R11: while we do not have many data on the topic in literature, we expanded the discussion section with the existing evidence, including a mention of the possible impact of the Italian public healthcare (see Discussion, page 8, line 266-280).

Reviewer 2 Report

In this article, Massarotti and colleagues reported the results of a study about healthy carriers of BRCA1/2 pathogenetic variants and the use of hormonal contraception and/or menopause hormonal therapy. In this study, including a big cohort of BRCA1/2 carriers (241), they highlighted that the majority of women (53.6% for contraception and 61.5% for menopause) reported being dissatisfied with the counselling received and 58.2% were not aware of the protective effect of hormonal contraception on the risk of ovarian cancer.  The authors concluded that the situation is worrisome and mandates a call to action for better education in this area. They highlighted that, while increasing attention is given to BRCA1/2 healthy carriers regarding risk-reducing surgery for cancer prevention, fertility preservation and pregnancy, at the moment it does not sufficiently highlight the importance of reliable contraception and evidence-based counselling after risk-reducing bilateral salpingo-oophorectomy.

I endorse the publication of this interesting and current paper in present form.

Author Response

We thank the Reviewer 2 for the kind words and we are happy to see interest about this important topic.